

# Amphibian diversity across three adjacent ecosystems in Área de Conservación Guanacaste, Costa Rica

Alex W. Edwards[1,*], Xavier A. Harrison[2,*], M. Alex Smith[3], Maria Marta Chavarría Díaz[4], Mahmood Sasa[5], Daniel H. Janzen[6], Winnie Hallwachs[6], Gerardo Chaves[5], Roberto Fernández[7], Caroline Palmer[1], Chloe Wilson[1], Alexandra North[1] and Robert Puschendorf[1]

[1] School of Biological and Marine Sciences, University of Plymouth, Plymouth, Devon, UK
[2] Centre for Ecology & Conservation, University of Exeter, Penryn, Cornwall, UK
[3] Department of Integrative Biology, University of Guelph, Guelph, Ontario, Canada
[4] Department of Research, Área de Conservación Guanacaste, Liberia, Guanacaste, Costa Rica
[5] School of Biology, Universidad de Costa Rica, San Pedro, San Jose, Costa Rica
[6] Department of Biology, University of Pennsylvania, Philadelphia, PA, United States of America
[7] Guanacaste Dry Forest Conservation Fund, Philadelphia, United States of America
* These authors contributed equally to this work.

Corresponding author
Robert Puschendorf,
robert.puschendorf@plymouth.ac.uk

## ABSTRACT

Amphibians are the most threatened species-rich vertebrate group, with species extinctions and population declines occurring globally, even in protected and seemingly pristine habitats. These 'enigmatic declines' are generated by climate change and infectious diseases. However, the consequences of these declines are undocumented as no baseline ecological data exists for most affected areas. Like other neotropical countries, Costa Rica, including Área de Conservación Guanacaste (ACG) in north-western Costa Rica, experienced rapid amphibian population declines and apparent extinctions during the past three decades. To delineate amphibian diversity patterns within ACG, a large-scale comparison of multiple sites and habitats was conducted. Distance and time constrained visual encounter surveys characterised species richness at five sites—Murciélago (dry forest), Santa Rosa (dry forest), Maritza (mid-elevation dry-rain forest intersect), San Gerardo (rainforest) and Cacao (cloud forest). Furthermore, species-richness patterns for Cacao were compared with historic data from 1987–8, before amphibians declined in the area. Rainforests had the highest species richness, with triple the species of their dry forest counterparts. A decline of 45% (20 to 11 species) in amphibian species richness was encountered when comparing historic and contemporary data for Cacao. Conservation efforts sometimes focus on increasing the resilience of protected areas, by increasing their range of ecosystems. In this sense ACG is unique containing many tropical ecosystems compressed in a small geographic space, all protected and recognised as a UNESCO world heritage site. It thus provides an extraordinary platform to understand changes, past and present, and the resilience of tropical ecosystems and assemblages, or lack thereof, to climate change.

## INTRODUCTION

Ongoing biodiversity loss and its associated impacts are a major global issue, with the current rate of extinctions unprecedented in recent time—over 1,000 times the probable natural background rate (*Barnosky et al., 2011*; *Ceballos, Ehrlich & Dirzo, 2017*; *Pimm et al., 2006*, *2014*). This loss of species is changing and impoverishing ecosystems all over the world (*Hooper et al., 2012*; *Pimm & Raven, 2000*; *Pimm et al., 1995*) and is a major concern for biologists and ecologists studying a wide range of taxa (*Ehrlich, 1995*, *Dirzo et al., 2014*, *Janzen & Hallwachs, 2020*, *Worm & Tittensor, 2011*), not to mention the tropical societies that are losing their natural wild capital. At the vanguard of this current extinction spasm however are amphibians, with more species threatened with extinction than any other major vertebrate taxon (*Stuart et al., 2004*).

Amphibian diversity is strongly correlated with environmental conditions such as precipitation, temperature, and available moisture. Available moisture can be measured as the relation between potential and actual evapotranspiration and appears to be a major determinant of amphibian diversity in Costa Rica (*Savage, 2002*), with extreme humid conditions (where precipitation greatly exceeds potential evapotranspiration) being associated with the highest diversity of species. Temperature is another essential driver of Costa Rican amphibian diversity and is reflected by changes in temperature along an altitudinal gradient—moving from cooler temperatures at higher elevations to warmer ones at lower elevations. For example, 65% of Costa Rican amphibians can be found within the premontane belt, potentially reflecting the overlap between the lower temperature limits of upland species and upper limits of lowland species (*Savage, 2002*). However, this means that individuals are highly susceptible to changes in these conditions (*Bickford et al., 2010*; *Ficetola & Maiorano, 2016*; *Ryan et al., 2015*; *Walls, Barichivich & Brown, 2013*), making them vulnerable to anthropogenic pressures.

There are approximately 8,480 known amphibian species (*Frost, 2022*), 41% of which are threatened with global extinction (*IUCN, 2018*) and 43% have declining populations (*Hof et al., 2011*; *Stuart et al., 2004*). Yet even these numbers are likely to be underestimated as our knowledge of tropical amphibian diversity and density is so poor (*Wake & Vredenburg, 2008*). It is widely agreed that amphibians face a constellation of threats, with many working synergistically to accelerate declines, including global climate change, habitat destruction and alteration, invasive species, overexploitation, and infectious disease (*Collins & Crump, 2009*). Amphibian population declines have been noted as early as the 1950s (*Houlahan et al., 2000*) but did not receive broad attention until the 1980s (although see *Alford, Dixon & Pechmann, 2001*), after several localities experienced rapid population crashes, with many of these occurring in seemingly pristine and protected areas (*Stuart et al., 2004*; *Burrowes, Joglar & Green, 2004*). These 'enigmatic' declines were thought to occur due to a myriad of factors (*Collins & Storfer, 2003*), but today two main causal factors have since been recognised: the pathogenic fungus *Batrachochytrium dendrobatidis* and climate change (*Blaustein & Dobson, 2006*; *Clare et al., 2016*; *Lips et al., 2006*, *2008*; *Pounds & Puschendorf, 2004*; *Pounds et al., 2006*; *Rohr et al., 2008*; *Whitfield et al., 2007*).

Similar to other regions in tropical Central America, declines of Costa Rican amphibians have occurred rapidly (within 2–3 years.) at elevations above 500 m (*Young et al., 2001*) and has resulted in the extirpation of endemics found at higher elevations (*Bolaños, 2002*; *Pounds et al., 1997*). Área de Conservación Guanacaste (ACG), which protects 120,000 ha of dry, rain and, cloud forest (and 43,000 ha of Pacific Ocean) in northwestern Costa Rica, (*Janzen, Hallwachs & Kappelle, 2016*) lost many amphibian species in the late 1980's, mostly in upland areas (*Puschendorf et al., 2019*).

Amphibian communities are already feeling the effects of climate change, both globally (*Blaustein et al., 2010*; *Corn, 2005*; *Li, Cohen & Rohr, 2013*) and within ACG. These impacts observed for amphibians are mirrored by other taxa, with many lowland ACG species of both vertebrates and invertebrates now being recorded at much higher elevations (*Smith, Hallwachs & Janzen, 2014*), whilst increased droughts have led to widespread tree and epiphyte mortality (*Powers et al., 2020*). Furthermore, *Janzen & Hallwachs (2021)* have witnessed a precipitous decline in insect numbers since they first started working in ACG since 1963 and 1978, respectively. This trend they attribute to climate change, specifically the expanded and irregular dry season in all three major ecosystems present in ACG. The evidence is mounting that climate change is not an abstract event that will impact the world and ACG in the future, but a catastrophe we are experiencing now. To understand the future impacts of climate change, it is important to know the species that are most at risk and their needs and characteristics.

To draw meaningful comparisons, document any potential shift in diversity and distribution of species and define and measure conservation targets, temporal baseline data is fundamental (*Mihoub et al., 2017*). Despite the well documented recent declines and extinctions of amphibians across the globe, baseline data for many tropical places is still scant (*Collen et al., 2008*; *Feeley & Silman, 2010*; *Siddig, 2019*). This well documented decline of tropical amphibian diversity is based on a limited number of localities in better studied countries such as Australia, Costa Rica, Panama, Ecuador and a few others (*Pounds & Crump, 1994*; *Richards, McDonald & Alford, 1994*; *Lips et al., 2006*; *Merino, Coloma & Almendáriz, 2006*). Most of these declines have occurred at higher elevations, but more recent work suggest lowland populations are not exempt, with declines tending to occur over longer time periods (*Whitfield et al., 2007*; *Ryan et al., 2014*). Despite Costa Rica being one of the better studied localities for amphibian declines, baseline data is still lacking for many important areas—including ACG.

Several studies have investigated amphibian species richness within ACG, but these tended to focus on a single forest type (*Bickford, 1994*; *Sasa & Solórzano, 1995*) and lacked population level data. Identifying long-term population trends is essential for any conservation endeavour but has proved difficult for most tropical amphibians due to the lack of historical baseline data and overall disinterest in gathering it. The few studies (*e.g.*, *Acosta-Chaves et al., 2019*; *Ryan et al., 2014*; *Whitfield et al., 2007*) that have incorporated long-term population data have found large-scale declines in amphibian populations. Over a 35-year period in the lowland rainforest of La Selva, Caribbean Costa Rica, *Whitfield et al. (2007)* documented a decline of 75% in terrestrial amphibian density since 1970. La Selva is a protected old-growth rainforest. Here we are building on these initial studies
and integrating abundance data in a large-scale comparison of several sites and habitats within ACG, providing vital baseline data valuable for understanding and anticipating long-term trends. Furthermore, by incorporating historic species richness data for one of the ACG cloud forest sites, where species richness declined in the late 1980's, we hypothesise that some species recovery should be noted, mirroring similar species re-discovery in many other sites in lower Central America, where declines occurred (*García-Rodríguez et al., 2012*; *Voyles et al., 2018*).

## METHODS

### Study sites

We sampled five sites in ACG which included: Cacao (10°55′36.264″N; 85°28′5.8794″W; 1,050 m above sea level (asl); cloud forest), San Gerardo (10°52′48″N; 85°23′20.3994″W; 573 m asl; rainforest), Maritza (10°57′727.0″N; 85°29′40.3″W; 590 m asl; mid-elevation dry-rain forest intersect), Murciélago (10°54′3.6354″N; 85°43′45.444″W; 80 m asl; dry forest) and Santa Rosa (10°50′16.7634″N; 85°37′7.2042″W; 289 m asl; dry forest; Fig. 1). All five sites are 4.5–37.5 km distance from each other. Murciélago has the highest mean annual temperature, whereas Cacao has the lowest (Table 1). Cacao has the highest mean annual precipitation and precipitation during the driest quarter, while Murciélago has the lowest annual precipitation (Table 1). Santa Rosa and Murciélago are comprised of a mosaic of relatively young dry forest in restoration from pastureland in the last three decades, with a few remaining tiny patches of older growth forest that escaped logging and burning. San Gerardo is a classical rainforest of 400–700 m elevation. Cacao and Maritza are both older forests, with a mix of old-growth and regenerating forests. Average annual rainfall at these study sites can vary and ranges between 1,613.3 ± 17.44 mm and 2,820 ± 56.35 mm (Mean ± SD; *Fick & Hijmans, 2017*) with a major part of this variation due to hurricane years. The mean annual temperature ranges between 20.74 °C ± 0.67 °C and 26.15 °C ± 0.18 °C (Mean ± SD; *Fick & Hijmans, 2017*), with a marked rainy season (May–December).

### Sampling methods

We collected data between the 09 August and 15 November 2017 (rainy season). At each site, 10 × 100 m long transects were established—split evenly between terrestrial and riparian habitats. Animals were captured within 2 m of the transect and extending 2 m in height. The distance between transects varied between 100 m and 4 km, depending on terrain and topography. We used distance and time constrained Visual Encounter Surveys (hereafter referred to as 'VES'; *Scott, 1994*; *von May et al., 2010*) for a duration of 40 min. We sampled three quarters of the transects at night (18:00–00:00 h) and the remainder during the day (10:20–15:30 h) to account for both diurnal and nocturnal species. We used VES as most amphibian species are nocturnal and previous studies have shown that VES's (*Crump & Scott, 1994*) are more effective than other methods when sampling at night (*Doan, 2003*; *Rödel & Ernst, 2004*) and have been shown to be of equal effectiveness to other methods when sampling for amphibians during the day (*Doan, 2003*). VES are an effective tool for detecting several salamander species of the Plethodontidae family
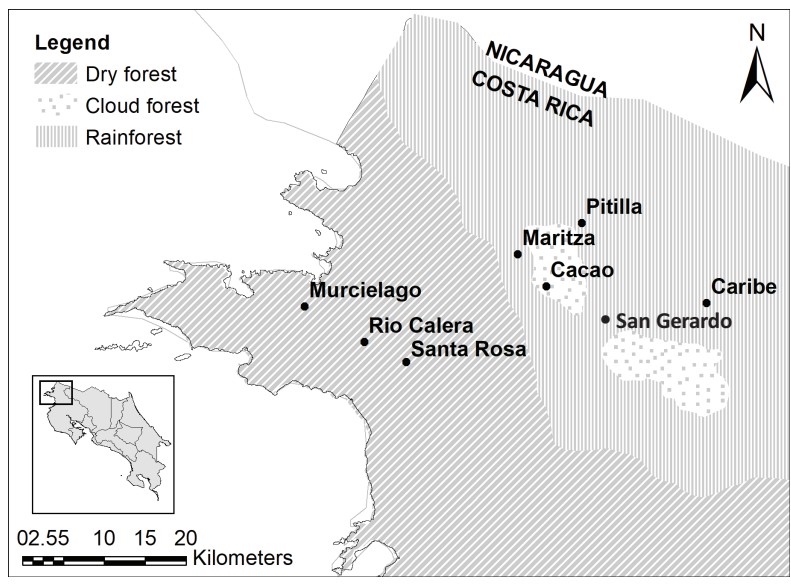

**Figure 1 Study sites.** Study sites in the Área de Conservación Guanacaste. Map was generated using open source data, from *Ortiz-Malavassi (2014)*. Figure Source: https://hdl.handle.net/2238/6749.

**Table 1 Climatic data for study sites.**

| Site | Annual mean temperature (°C) | Annual precipitation (mm) | Precipitation of the driest quarter (mm) | Elevation (m asl) |
|---|---|---|---|---|
| Cacao | 20.74 | 2,820.00 | 154.39 | 1,050 |
| San Gerardo | 23.15 | 2,558.18 | 104.45 | 573 |
| Maritza | 22.93 | 2,599.29 | 106.29 | 590 |
| Murciélago | 26.15 | 1,613.30 | 15.50 | 80 |
| Santa Rosa | 24.76 | 1,700.90 | 11.70 | 289 |
| All sites | 23.17 | 2,325.66 | 88.29 | 516 |

**Note:**
Climatic data for each of five sites and all sites pooled together. The environmental envelope for each site was extracted at a 1 km$^2$ resolution from WorldClim.

(*Grover, 2006*), however species in the genera *Nototriton* and *Oedipina* are best sampled using cover object searches which can damage fragile habitats—notably mosses and bromeliads. No specific efforts were therefore taken to conduct destructive sampling of a fragile cloud forest ecosystem in search of salamanders.

On terrestrial transects amphibians located 2 m either side of the transect centre were captured, for a total width of 4 m and on substrates up to 2 m in height (*von May & Donnelly, 2009*). Captured individuals were placed in their own plastic bags with substrate and water for moisture and labelled with a unique identification code and location on the transect. Further biosecurity precautions (*e.g.*, new gloves for each capture) were deemed unnecessary due to the high prevalence of *B. dendrobatidis* and *Ranavirus* within the ACG (*Wynne, 2018*; *Puschendorf et al., 2019*). Most individuals were released at the end of the survey, but some were brought back for further identification and released the next day back at the point of capture.

We resampled transects at 2-to-4-day intervals, with each transect sampled four times during this study. After the transect was set up a minimum of 2 days were left before surveying began, to minimise any impact from disturbance on sampling. We measured and marked down every 10 m on transects using flagging tape which we collected at the end of the study. GPS coordinates and elevation were collected at the midpoint of each transect using a Garmin 60CSX. Annual mean temperature, annual precipitation and precipitation of the driest quarter were extracted for each field site from WorldClim (version 1.4) at a 1 km$^2$ resolution (*Hijmans et al., 2005*).

Historic data for Cacao was obtained from Arctos Collaborative Collection (*MVZ, 2018*) management solutions museum database. Data were collected by David Cannatella and David Good over 23 days between July 1987 and January 1988—with most sampling occurring in August 1987 (For species list see Table S1). There was no standardised sampling, observers walked through the forest collecting everything they came across (D. Cannatella, 2018, personal communication). Historic data for Cacao is hereafter referred to as historic Cacao. This work was carried out under CONAGEBIO Permit number R-036-2013-OT-CONAGEBIO.

## Data analysis

Unless otherwise stated, all statistical analysis was conducted in the R statistical environment v4.1.2 (*R Core Team, 2022*). We used the numbers equivalent approach as suggested by *Jost (2006, 2007)* to describe patterns of beta diversity and community similarity across sites using the package 'vegetarian' (*Charney & Record, 2012*). β-diversity was analysed based on the numbers equivalent of Shannon's diversity $^1D_\beta$ using the diversity order q = 1 which considers the proportional abundance of each species in a community, without favouring either rare or abundant species (*Jost, 2006*). Ten thousand bootstrap replicates of the data were used to determine standard error of β-diversity for each site.

We performed sample-based rarefaction analyses to compare patterns of species richness between sites (*Gotelli & Colwell, 2001*). Transect data were pooled across sites and the 'vegan' package (*Oksanen et al., 2007*) was used to generate the subsequent comparisons. A sample-based rarefaction curve was further used to compare species richness patterns between historic and current data for Cacao.

To estimate inventories completeness, we used the nonparametric estimators of species richness; ACE and Chao1 based on abundance data (*Hughes et al., 2001*; *Jiménez-Valverde & Hortal, 2003*), using EstimateS Program V9.1.0 (*Chao, 1984*; *Chao & Lee, 1992*; *Chao & Yang, 1993*; *Chazdon et al., 1998*; *Colwell, 2013*; *Colwell & Coddington, 1994*).

To compare species abundance patterns between sites, rank abundance curves (RAC) were plotted (*Magurran, 2004*) using the BiodiversityR package (*Kindt & Coe, 2005*). The slope of linear regression of an RAC expresses the evenness in abundance among species within an assemblage and an analysis of covariance (ANCOVA) was used to compare differences in evenness among sites. An abundant species was arbitrarily defined as those that were represented by more than 12 individuals (which is approximately 2% of all individuals across the study). We used the package brms (*Bürkner, 2017, 2018*) to test

for differences among sites in the rate of decay in rank abundance slopes. We specified per-species abundance as an outcome variable, with a negative binomial error structure. We included the interaction between rank and site as fixed effects, allowing the slope of decay to vary by site. We assessed model fit using visual inspection of mcmc chains, and posterior predictive checks. We determined differences between sites in rates of abundance decay based on whether differences in 95% credible intervals of slope parameters included zero. We used the Leave One Out Information Criterion (LOO-IC, *Vehtari, Gelman & Gabry, 2017*; *Vehtari et al., 2020*) to perform a full model test of the maximal model against the intercept only model (*Forstmeier & Schielzeth, 2011*).

Multidimensional scaling (nMDS) ordination (k = 2, stress = 0.12) using the 'vegan' package (*Oksanen et al., 2022*) was used to visualise the difference in community structure and composition among sites. The nMDS plot is based on a Jaccard matrix, using species presence/absence data. Additionally, the similarity percentage (SIMPER: *Clarke & Warwick, 2001*) was calculated to identify the contribution of individual species to the dissimilarity of amphibian community structure among sites. Moreover, a SIMPER analysis was also conducted using the historic data for Cacao to understand the changes in community structure over time and how this has affected inter-site relatedness. Abundance was analysed after a square root transformation of the data. This was conducted using the 'vegan' package (*Oksanen et al., 2007*).

All code and datasets required for reproducing these results, including model fitting and data visualisation, are provided online (https://github.com/xavharrison/CostaRica_RankAbundance_2022).

## RESULTS

During the surveys between August–November 2017, 660 individual amphibians from 37 species were recorded, all anurans, (Table S2). This represents 46.25% of known amphibian species to occur in ACG (Table S3). The overall sampling effort was 267 person-hours throughout the entire study. In total 50 transects were resampled four times for a total of 200 transects. Several other individuals and species were captured outside of standard sampling (Table S4), but those have not been included in this analysis. *Duellmanohyla rufioculis*, *Craugastor fitzingeri*, *Rhaebo haematiticus* and *Craugastor crassidigitus* were the most common species, comprising 20.3%, 13.5%, 11.7% and 11.7% of the total captured. We recorded nine amphibian families (all anuran), with three families represented by only a single species: Microhylidae (*Hypopachus variolosus*), Phyllomedusidae (*Agalychnis callidryas*) and Eleutherodactylidae (*Diasporus diastema*).

All sites had low similarity based upon species abundance (Horn index $\pm$ *SD*: 0.19 $\pm$ 0.17). The overall β-diversity for all sites combined was 3.16 $\pm$ 0.134 ($^1D_\beta \pm SD$), highest in San Gerardo ($^1D_\beta = 3.27 \pm 0.26$) and lowest in Santa Rosa ($^1D_\beta = 1.23 \pm 0.11$). β-diversity for the remaining sites was as follows; Cacao ($^1D_\beta = 2.02 \pm 0.11$), Maritza ($^1D_\beta = 1.98 \pm 0.16$) and, Murciélago ($^1D_\beta = 2.14 \pm 0.20$).

The sample size was sufficient to characterise species richness for three of the five sites; Cacao; San Gerardo and Santa Rosa, as the rarefaction curve approaches an asymptote (Fig. 2A). The highest number of species was recorded in San Gerardo (rainforest) and the

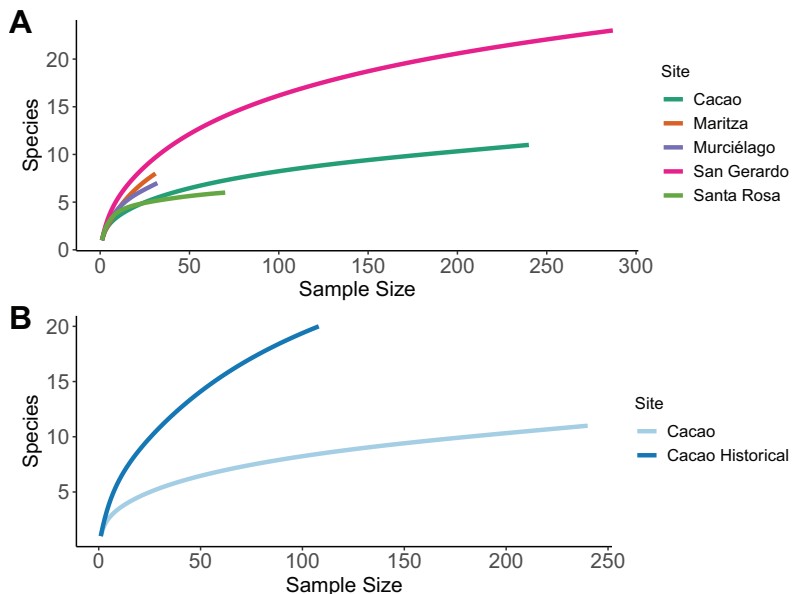

**Figure 2 Sample-based rarefaction curves.** Sample-based rarefaction curves, whereby each curve represents the expected number of species for a certain number of observed individuals. (A) Comparison among five different sites within the ACG, representing the four main forest ecosystems. (C) Comparison among historical (1987/8) and current (2017) data for one one sites (Cacao).

least in Santa Rosa. In Cacao, a total of 20 species were recorded in the 1980's compared to only 11 in 2017, a decline of 45% (Fig. 2B). Of the 11-species recorded in 2017, three of them were absent from the 1987 data—*Craugastor fitzingeri*, *Hyalinobatrachium colymbiphyllum* and *Smilisca baudinii*. Furthermore, the curve for the historic data failed to reach an asymptote, suggesting that the inventory was incomplete at that stage and more species remained to be discovered. This is supported by museum records and data collected and stored at Arctos Collaborative Collection management solutions (*MVZ, 2018*), which suggest a total of 39 species are known to occur in Cacao (Table S5).

Overall estimates of completeness were highest for Santa Rosa (ACE = 85.71% and Chao1 = 100%) and San Gerardo, which was predicted to be missing seven species (Table 2). Cacao had the lowest level of completeness (ACE = 68.75% and Chao1 = 64.71%), as 54.58% of all individuals encountered were *Duellmanohyla rufioculis*.

Our Bayesian regression (Table 3), suggests that Cacao was found to have much higher species abundances at lower ranks. Whilst all sites decayed at a similar rate (*i.e.* had similar slopes), the site:rank interaction in the model revealed San Gerardo to have a much shallower rate of decline (Figs. 3 and S1). Low density species (represented by a single individual) also mainly occurred in San Gerardo as well as Cacao. The abundance distribution in Murciélago and Santa Rosa suggests that these sites today have less abundant species as compared with San Gerardo (Fig. 4). *Rhinella horribilis* was the most dominant species in both Murciélago and Santa Rosa. In contrast the dominant species in Cacao and Maritza (*Duellmanohyla rufioculis* and *Lithobates warszewitschii*) are not found in lowland sites (*Savage, 2002*).

**Table 2 Observed species richness.**

| Site | $S_{obs}$ | Species predicted ACE | % Of completeness ACE | Species predicted Chao1 | % Of completeness Chao1 |
|---|---|---|---|---|---|
| Cacao | 11 | 16 | 68.75 | 17 | 64.71 |
| San Gerardo | 23 | 30 | 76.67 | 30 | 76.67 |
| Maritza | 8 | 12 | 66.67 | 11 | 72.73 |
| Murciélago | 7 | 10 | 70 | 10 | 70 |
| Santa Rosa | 6 | 7 | 85.71 | 6 | 100 |

**Note:**
Observed species richness (Sobs), number of species predicted by the nonparametric species richness estimators ACE and Chao1, and the percentage completeness of each site based on these estimators.

**Table 3 Model estimates.**

| | Mean | Lower 95% CI | Upper 95% CI |
|---|---|---|---|
| Intercept (Cacao) | 5.27 | 4.75 | 5.78 |
| Rank | **−0.64** | **−0.78** | **−0.51** |
| Maritza | **−2.21** | **−3.16** | **−1.26** |
| Murciélago | **−2.09** | **−3.05** | **−1.09** |
| SanGerardo | **−0.94** | **−1.59** | **−0.35** |
| SantaRosa | **−1.16** | **−2.07** | **−0.15** |
| Rank: Maritza | 0.1 | −0.2 | 0.38 |
| Rank: Murciélago | 0.07 | −0.24 | 0.36 |
| Rank: SanGerardo | **0.39** | **0.25** | **0.53** |
| Rank: SantaRosa | 0.04 | −0.26 | 0.32 |
| $R^2$ | 0.83 | 0.61 | 0.95 |

**Note:**
Model estimates from best supported model containing means and 95% credible intervals are shown in bold.

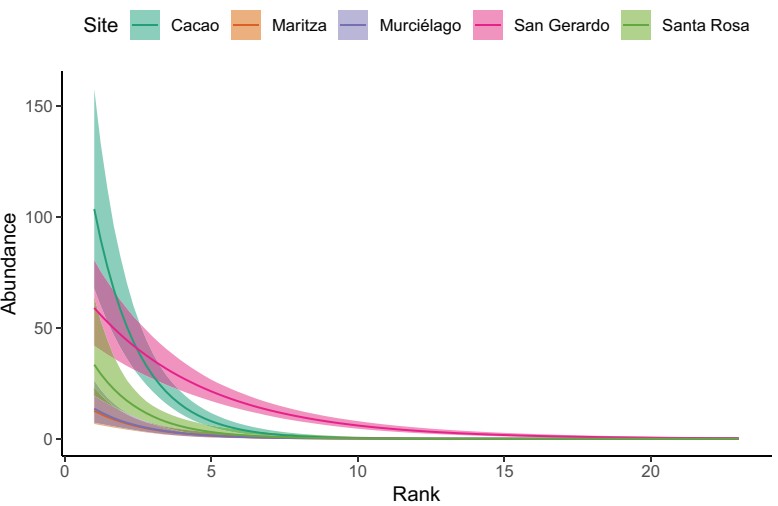

**Figure 3 Site rank decay curves for the five sampled sites.** Site rank decay curves for the five sampled sites. Bold lines represent posterior means, and shaded areas are 95% credible intervals from a negative binomial GLM.

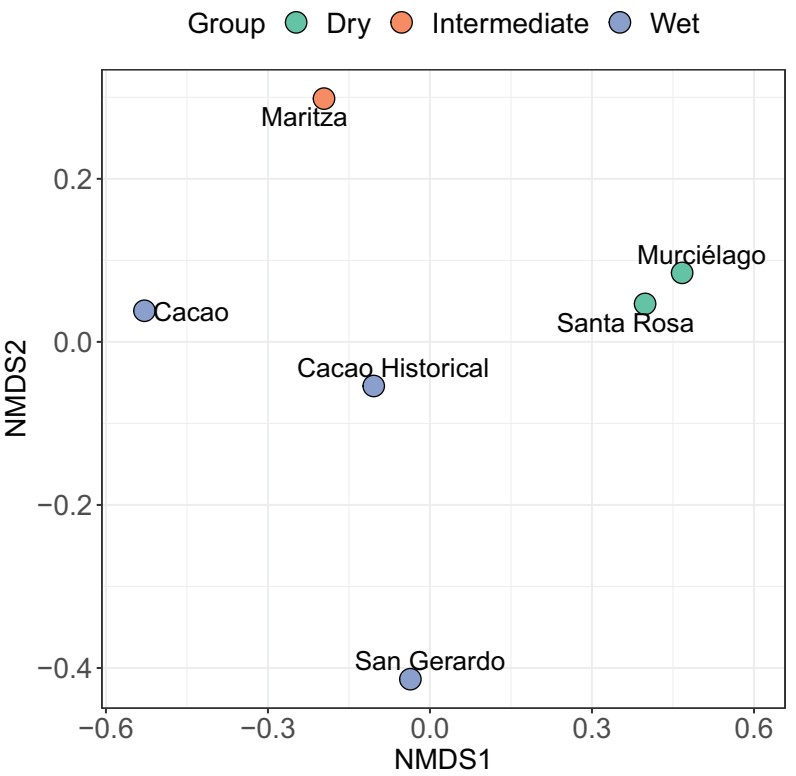

**Figure 4 Non-metric multidimensional scaling (NMDS) ordination.** Results of a non-metric multi-dimensional scaling (NMDS) ordination for six amphibian communities across Área de Conservación Guanacaste (ACG). Solid symbols indicate the site and habitat type of individual communities (blue = wet forest, red = dry forest, and green = mid-elevation dry-rain intersect). Abundance data was transformed by dividing each value by the row total (a simple transformation when some sites have higher abundance than others). The NMDS1 axis can be seen as a measure of temperature and precipitation while the NMDS2 axis is a measure of the seasonality of precipitation (calculated using vectors of BIOCLIM variables).

The nMDS shows a clear split between most of the sites. Santa Rosa and Murciélago are the most similar sites, followed by Cacao and Cacao historic (Fig. 4). Excluding Cacao historic, San Gerardo was identified as the most unique site, but this was closely followed by Maritza. However, including Cacao historic resulted in Maritza being the most unique. The SIMPER analysis suggests that the community structure of the five sites is distinct from each other, despite the short geographic distance between them (Table 4), with an average dissimilarity of 83.20%. Murciélago and Santa Rosa were the least dissimilar sites, with a dissimilarity of 60.97%, followed by Cacao and Maritza with a dissimilarity of 71.66%. Cacao and Santa Rosa had the highest dissimilarity between sites at 96.37%. The SIMPER analysis using the historic data for Cacao showed an increase in similarity between Cacao and the other sites over the 30-year period (1987/8–2017). As expected, the historic data for Cacao was most like contemporary Cacao, with a dissimilarity of 61.87%. All sites, except Santa Rosa, experienced an increase in similarity between the two periods with Maritza experiencing the biggest drop, with a decrease in dissimilarity from 81.77% to 71.66%. The dissimilarity between Santa Rosa and Cacao increased between the two sampling periods, increasing from 75.35% to 96.37%.

**Table 4 SIMPER analysis, showing dissimilarity between sites based on amphibian assemblages.**

| Comparison | Overall diss. (%) | Species* | Average abundance | | Average diss. | Contribution (%) | Cumulative contribution (%)** |
|---|---|---|---|---|---|---|---|
| Cacao v. San Gerardo | 76.05 | | Cac | S G | | | |
| | | D. rufioculis | 11.45 | 0.00 | 11.62 | 15.27 | 15.27 |
| | | R. haematiticus | 0.00 | 8.77 | 8.91 | 11.71 | 26.98 |
| | | C. fitzingeri | 2.00 | 9.11 | 7.22 | 9.49 | 36.47 |
| | | C. megacephalus | 1.73 | 6.32 | 4.66 | 6.13 | 42.6 |
| | | C. crassidigitus | 8.00 | 3.61 | 4.46 | 5.86 | 48.47 |
| | | T. pulverata | 0.00 | 3.87 | 3.93 | 5.17 | 53.64 |
| | | L. vaillanti | 0.00 | 3.00 | 3.04 | 4.00 | 57.64 |
| | | L. warszewitschii | 4.47 | 1.73 | 2.78 | 3.66 | 61.3 |
| | | S. sordida | 0.00 | 2.65 | 2.69 | 3.53 | 64.83 |
| | | C. bransfordii | 0.00 | 2.24 | 2.27 | 2.98 | 67.81 |
| | | T. spinosa | 0.00 | 2.24 | 2.27 | 2.98 | 70.79 |
| Cacao v. Maritza | 71.66 | | Cac | Mar | | | |
| | | D. rufioculis | 11.45 | 1.73 | 19.41 | 27.08 | 27.08 |
| | | C. crassidigitus | 8.00 | 0.00 | 15.98 | 22.3 | 49.39 |
| | | P.ridens | 3.16 | 0.00 | 6.32 | 8.82 | 58.2 |
| | | H. colymbiphyllum | 2.00 | 0.00 | 4 | 5.58 | 63.78 |
| | | T. typhonius | 0.00 | 1.73 | 3.46 | 4.83 | 68.61 |
| | | C. megacephalus | 1.73 | 0.00 | 3.46 | 4.83 | 73.44 |
| Cacao v. Murciélago | 91.98 | | Cac | Mur | | | |
| | | D. rufioculis | 11.45 | 0.00 | 22.94 | 24.93 | 24.93 |
| | | C. crassidigitus | 8.00 | 0.00 | 16.03 | 17.43 | 42.36 |
| | | L. warszewitschii | 4.47 | 0.00 | 8.96 | 9.74 | 52.1 |
| | | R. horribilis | 0.00 | 4.12 | 8.26 | 8.98 | 61.09 |
| | | P. ridens | 3.16 | 0.00 | 6.34 | 6.89 | 67.98 |
| | | C. ranoides | 0.00 | 2.24 | 4.48 | 4.87 | 72.85 |
| Cacao v. Santa Rosa | 96.37 | | Cac | S R | | | |
| | | D. rufioculis | 11.45 | 0.00 | 20.77 | 21.55 | 21.55 |
| | | C. crassidigitus | 8.00 | 0.00 | 14.52 | 15.07 | 36.62 |
| | | R. horribilis | 0.00 | 5.57 | 10.10 | 10.49 | 47.11 |
| | | L. warszewitschii | 4.47 | 0.00 | 8.12 | 8.42 | 55.53 |
| | | E. pustulosus | 0.00 | 3.87 | 7.03 | 7.29 | 62.82 |
| | | P. ridens | 3.16 | 0.00 | 5.74 | 5.96 | 68.78 |
| | | H. variolosus | 0.00 | 2.83 | 5.13 | 5.33 | 74.10 |
| Cacao v. Cacao (Historical) | 61.87 | | Cac | Cac (H) | | | |
| | | C. crassidigitus | 8.00 | 1.73 | 8.30 | 13.42 | 13.42 |
| | | D. rufioculis | 11.45 | 5.20 | 8.28 | 13.38 | 26.80 |
| | | N. guanacaste | 0.00 | 4.24 | 5.62 | 9.08 | 35.89 |
| | | L. forreri | 1.00 | 4.24 | 4.30 | 6.94 | 42.83 |
| | | L. warszewitschii | 4.47 | 2.00 | 3.27 | 5.29 | 48.12 |

*(Continued)*

 

| Table 4 (continued) | | | | | | |
|---|---|---|---|---|---|---|
| **Comparison** | **Overall diss. (%)** | **Species*** | **Average abundance** | | **Average diss.** | **Contribution (%)** | **Cumulative contribution (%)*** |

| Comparison | Overall diss. (%) | Species* | Average abundance | | Average diss. | Contribution (%) | Cumulative contribution (%)** |
|---|---|---|---|---|---|---|---|
| | | *C. megacephalus* | 1.73 | 3.87 | 2.84 | 4.58 | 52.71 |
| | | *L. forreri* | 2.00 | 0.00 | 2.65 | 4.28 | 56.99 |
| | | *C. fitzingeri* | 2.00 | 0.00 | 2.65 | 4.28 | 61.27 |
| | | *H. colymbiphyllum* | 0.00 | 2.00 | 2.65 | 4.28 | 65.55 |
| | | *C. fitzingeri* | 3.16 | 1.73 | 1.89 | 3.06 | 68.62 |
| | | *C. melanostictus* | 0.00 | 1.41 | 1.87 | 3.03 | 71.64 |
| | | *P. ridens* | | | | | |
| | | *H. variolosus* | | | | | |
| San Gerardo v. Maritza | 90.04 | | S G | Mar | | | |
| | | *R. haematiticus* | 8.77 | 0.00 | 11.71 | 13.00 | 13.00 |
| | | *C. fitzingeri* | 9.11 | 1.00 | 10.82 | 12.02 | 25.02 |
| | | *C. megacephalus* | 6.32 | 0.00 | 8.44 | 9.37 | 34.39 |
| | | *T. pulverata* | 3.87 | 0.00 | 5.17 | 5.74 | 40.12 |
| | | *C. crassidigitus* | 3.61 | 0.00 | 4.81 | 5.34 | 45.47 |
| | | *S. baudinii* | 3.16 | 0.00 | 4.22 | 4.68 | 50.15 |
| | | *L. vaillanti* | 3.00 | 0.00 | 4.00 | 4.44 | 54.60 |
| | | *S. sordida* | 2.65 | 0.00 | 3.53 | 3.92 | 58.51 |
| | | *L. warszewitschii* | 1.73 | 4.36 | 3.50 | 3.89 | 62.41 |
| | | *C. bransfordii* | 2.24 | 0.00 | 2.98 | 3.31 | 65.72 |
| | | *T. spinosa* | 2.24 | 0.00 | 2.98 | 3.31 | 69.03 |
| | | *T. typhonius* | 0.00 | 1.73 | 2.31 | 2.57 | 71.60 |
| San Gerardo v. Murciélago | 87.35 | | S G | Mur | | | |
| | | *R. haematiticus* | 8.77 | 0.00 | 11.73 | 13.43 | 13.43 |
| | | *C. fitzingeri* | 9.11 | 1.00 | 10.84 | 12.41 | 25.84 |
| | | *C. megacephalus* | 6.32 | 0.00 | 8.45 | 9.68 | 35.52 |
| | | *T. pulverata* | 3.87 | 0.00 | 5.18 | 5.93 | 41.44 |
| | | *C. crassidigitus* | 3.61 | 0.00 | 4.82 | 5.52 | 46.96 |
| | | *L. vallianti* | 3.00 | 0.00 | 4.01 | 4.59 | 51.55 |
| | | *S. sordida* | 2.65 | 0.00 | 3.54 | 4.05 | 55.6 |
| | | *R. horribilis* | 1.73 | 4.12 | 3.2 | 3.66 | 59.26 |
| | | *C. ranoides* | 0.00 | 2.24 | 2.99 | 3.42 | 62.68 |
| | | *C. bransfordii* | 2.24 | 0.00 | 2.99 | 3.42 | 66.1 |
| | | *T. spinosa* | 2.24 | 0.00 | 2.99 | 3.42 | 69.52 |
| | | *I. leutkenii* | 0.00 | 1.73 | 2.32 | 2.65 | 72.17 |
| San Gerardo v. Santa Rosa | 95.67 | | S G | S R | | | |
| | | *C. fitzingeri* | 9.11 | 0.00 | 11.39 | 11.9 | 11.9 |
| | | *R. haematiticus* | 8.77 | 0.00 | 10.97 | 11.46 | 23.36 |
| | | *C. megacephalus* | 6.32 | 0.00 | 7.9 | 8.26 | 31.63 |
| | | *E. pustulosus* | 0.00 | 3.87 | 4.84 | 5.06 | 36.69 |
| | | *T. pulverata* | 3.87 | 0.00 | 4.84 | 5.06 | 41.75 |

| Comparison | Overall diss. (%) | Species* | Average abundance | | Average diss. | Contribution (%) | Cumulative contribution (%)** |
|---|---|---|---|---|---|---|---|
| | | | | | | | |
| | | R. horribilis | 1.73 | 5.57 | 4.79 | 5.01 | 46.76 |
| | | L. forreri | 0.00 | 3.61 | 4.51 | 4.71 | 51.47 |
| | | C. crassidigitus | 3.61 | 0.00 | 4.51 | 4.71 | 56.18 |
| | | S. baudinii | 3.16 | 0.00 | 3.95 | 4.13 | 60.31 |
| | | L. vaillanti | 3.00 | 0.00 | 3.75 | 3.92 | 64.23 |
| | | H. variolosus | 0.00 | 2.83 | 3.53 | 3.69 | 67.92 |
| | | S. sordida | 2.65 | 0.00 | 3.31 | 3.46 | 71.38 |
| San Gerardo v. Cacao (Historical) | 77.12 | | S G | Cac (H) | | | |
| | | C. fitzingeri | 9.11 | 0.00 | 9.07 | 11.77 | 11.77 |
| | | R. haematiticus | 8.77 | 0.00 | 8.74 | 11.33 | 23.1 |
| | | D. rufioculis | 0.00 | 5.20 | 5.18 | 6.71 | 29.81 |
| | | L. forreri | 1.41 | | 4.23 | 6.06 | 35.87 |
| | | N. guanacaste | 0.00 | 4.24 | 4.23 | 5.00 | 40.87 |
| | | T. pulverata | 0.00 | 4.24 | 3.86 | 4.90 | 45.77 |
| | | S. baudinii | 3.87 | 0.00 | 3.15 | 4.08 | 49.85 |
| | | L. vaillanti | 3.16 | 0.00 | 2.99 | 3.87 | 53.73 |
| | | S. sordida | 3.61 | | 2.64 | 3.42 | 57.14 |
| | | C. megacephalus | 3.00 | 0.00 | 2.44 | 3.17 | 60.31 |
| | | T. spinosa | 2.65 | 0.00 | 2.23 | 2.89 | 63.20 |
| | | C. melanostictus | 6.32 | 3.87 | 1.99 | 2.58 | 65.78 |
| | | C. crassidigitus | 2.24 | 0.00 | 1.87 | 2.42 | 68.20 |
| | | C. persimilis | 0.00 | 2.00 | 1.73 | 2.24 | 70.44 |
| | | | 3.61 | 1.73 | | | |
| | | | 1.73 | 0.00 | | | |
| Maritza v. Murciélago | 77.21 | | Mar | Mur | | | |
| | | L. warszewitschii | 4.36 | 0.00 | 16.56 | 21.44 | 21.44 |
| | | R. horribilis | 1.00 | 4.12 | 11.86 | 15.36 | 36.81 |
| | | C. ranoides | 0.00 | 2.24 | 8.49 | 11 | 47.81 |
| | | S. baudinii | 0.00 | 2.00 | 7.6 | 9.84 | 57.64 |
| | | I. leutkenii | 0.00 | 1.73 | 6.58 | 8.52 | 66.16 |
| | | D. rufioculis | 1.73 | 0.00 | 6.58 | 8.53 | 74.68 |
| Maritza v. Santa Rosa | 84.68 | | Mar | S R | | | |
| | | R. horribilis | 1.00 | 5.57 | 14.49 | 17.11 | 17.11 |
| | | L. warszewitschii | 4.36 | 0.00 | 13.83 | 16.33 | 33.44 |
| | | E. pustulosus | 0.00 | 3.87 | 12.28 | 14.51 | 47.94 |
| | | L. forreri | 0.00 | 3.61 | 11.44 | 13.51 | 61.45 |
| | | H. variolosus | 0.00 | 2.83 | 8.97 | 10.59 | 72.04 |

(Continued)

| Comparison | Overall diss. (%) | Species* | Average abundance | | Average diss. | Contribution (%) | Cumulative contribution (%)** |
|---|---|---|---|---|---|---|---|
| Maritza v. Cacao (Historical) | 81.77 | | Mar | Cac (H) | | | |
| | | L. forreri | 0.00 | 4.24 | 8.17 | 9.99 | 9.99 |
| | | N. guanacaste | 0.00 | 4.24 | 8.17 | 9.99 | 19.99 |
| | | C. megacephalus | 0.00 | 3.87 | 7.46 | 9.12 | 29.11 |
| | | D. rufioculis | 1.73 | 5.20 | 6.67 | 8.16 | 37.27 |
| | | L. warszewitschii | 4.36 | 2.00 | 4.54 | 5.56 | 42.83 |
| | | C. melanostictus | 0.00 | 2.00 | 3.85 | 4.71 | 47.54 |
| | | T. typhonius | 1.73 | 0.00 | 3.34 | 4.08 | 51.62 |
| | | C. crassidigitus | 0.00 | 1.73 | 3.34 | 4.08 | 55.7 |
| | | P. ridens | 0.00 | 1.73 | 3.34 | 4.08 | 59.78 |
| | | H.variolosus | 0.00 | 1.41 | 2.72 | 3.33 | 63.12 |
| | | C. bransfordii | 0.00 | 1.41 | 2.72 | 3.33 | 66.45 |
| | | D. diastema | 0.00 | 1.41 | 2.72 | 3.33 | 69.78 |
| | | C. granulisa | 1.41 | 0.00 | 2.72 | 3.33 | 73.11 |
| Murciélago v. Santa Rosa | 60.97 | | Mur | S R | | | |
| | | L. forreri | 0.00 | 3.61 | 11.49 | 18.84 | 18.84 |
| | | E. pustulosus | 1.00 | 3.87 | 9.16 | 15.02 | 33.86 |
| | | H. variolosus | 0.00 | 2.83 | 9.01 | 14.78 | 48.64 |
| | | C. ranoides | 2.24 | 0.00 | 7.13 | 11.69 | 60.33 |
| | | S. baudinii | 2.00 | 0.00 | 6.37 | 10.45 | 70.78 |
| Murciélago v. Cacao (Historical) | 92.27 | | Mur | Cac (H) | | | |
| | | D. rufioculis | 0.00 | 5.20 | 10.04 | 10.88 | 10.88 |
| | | L. forreri | 0.00 | 4.24 | 8.20 | 8.88 | 19.76 |
| | | N. guanacaste | 0.00 | 4.24 | 8.20 | 8.88 | 28.64 |
| | | C. megacephalus | 0.00 | 3.87 | 7.48 | 8.18 | 36.75 |
| | | R. horribilis | 4.12 | 1.00 | 6.03 | 6.54 | 43.29 |
| | | C. ranoides | 2.24 | 0.00 | 4.32 | 4.68 | 47.97 |
| | | S. baudinii | 2.00 | 0.00 | 3.86 | 4.19 | 52.16 |
| | | L. warszewitschii | 0.00 | 2.00 | 3.86 | 4.19 | 56.34 |
| | | C. melanosticus | 0.00 | 2.00 | 3.86 | 4.19 | 60.53 |
| | | I. leutkenii | 1.73 | 0.00 | 3.35 | 3.63 | 64.16 |
| | | C. crassidigitus | 0.00 | 1.73 | 3.35 | 3.63 | 67.78 |
| | | P. ridens | 0.00 | 1.73 | 3.35 | 3.63 | 71.41 |
| Santa Rosa v. Cacao (Historical) | 75.35 | | S R | Cac (H) | | | |
| | | D. rufioculis | 0.00 | 5.20 | 9.12 | 12.11 | 12.11 |
| | | R. horribilis | 5.57 | 1.00 | 8.02 | 10.64 | 22.75 |
| | | N. guanacaste | 0.00 | 4.24 | 7.45 | 9.88 | 32.63 |
| | | C. megacephalus | 0.00 | 3.87 | 6.80 | 9.02 | 41.65 |
| | | E. pustulosus | 3.87 | 1.00 | 5.04 | 6.69 | 48.35 |
| | | L. warszewitschii | 0.00 | 2.00 | 3.51 | 4.66 | 53.01 |

| Comparison | Overall diss. (%) | Species* | Average abundance | | Average diss. | Contribution (%) | Cumulative contribution (%)** |
|---|---|---|---|---|---|---|---|
| | | *C. melanostictus* | 0.00 | 2.00 | 3.51 | 4.66 | 57.66 |
| | | *C. crassidigitus* | 0.00 | 1.73 | 3.04 | 4.04 | 61.70 |
| | | *P. ridens* | 0.00 | 1.73 | 3.04 | 4.04 | 65.74 |
| | | *H. variolosus* | 2.83 | 1.41 | 2.48 | 3.29 | 69.03 |
| | | *T. typhonius* | 1.41 | 0.00 | 2.48 | 3.29 | 72.32 |

**Notes:**
* The species contributing the most towards distinguishing between the habitats.
** Contributions of all species until a threshold of 70% of the total dissimilarity between groups is explained.

# DISCUSSION

Our analysis presented here reinforces that amphibian species richness is strongly correlated with forest type. This pattern follows the diverging environmental conditions present in each forest type, which has resulted in very different communities across ACG. Furthermore, we observed a substantial decrease in amphibian species richness over time, at the relatively undisturbed cloud forest site Cacao. This is further evidence for the widespread decline of amphibians observed globally and in Costa Rica over the past several decades, and recovery is still tenuous, if at all (*Lips et al., 2006*; *Stuart et al., 2004*; *Whitfield, Lips & Donnelly, 2016*).

Historic museum records kept at the Museo de Zoología, Universida de Costa Rica have documented 80 species, consisting of 75 Anurans, one Gymnophiona and four Caudata within ACG. We detected 37 species of anurans in the three main ACG ecosystems. Many ACG areas have yet to be surveyed more than superficially; and will contain unrecorded or new species. For example, during the pilot study we discovered *Agalychnis saltator* in Pitilla (Table S4), which represents a substantial range expansion for this species and a species new to ACG. Furthermore, new molecular approaches are revealing previously undescribed amphibian cryptic diversity (*Funk, Caminer & Ron, 2012*; *Stuart, Inger & Voris, 2006*) including in ACG frogs (*e.g.*, *Cryer et al., 2019*). Finally, sampling across seasons and years will be key to elucidating the full diversity of ACG amphibians, with many species experiencing yearly fluctuations in population size (*Marsh, 2001*) and higher visibility in specific seasons (*Laurencio & Fitzgerald, 2010*, *Savage, 2002*).

Rainforests had the highest levels of amphibian species richness, which support previous findings for Costa Rica (*Savage, 2002*) and elsewhere (*Duellman & Trueb, 1994*). The three forest types sampled are in part defined by their evolutionary history, vegetation communities, previous disturbance and stage of restoration, levels of precipitation, temperature and the annual actual evapotranspiration (AET; *Janzen, Hallwachs & Kappelle, 2016*). It has been demonstrated that a mixture of water and energy variables are important in shaping amphibian species richness patterns in North America, Europe, Asia and Central America (*Currie, 2001*; *Laurencio & Fitzgerald, 2010*; *Rodríguez, Belmontes & Hawkins, 2005*). For example, *Qian et al. (2017)* found a strong positive correlation between amphibian species richness and environmental variables such as precipitation, net

primary productivity, range in elevation and temperature; in 245 localities across China. These findings demonstrate that environmental variables may play a role in constraining the species richness at a site and constitutes the most plausible explanation for the differences between the forest types. This is supported by the fact that dry forest sites, prior to disturbance, had much lower levels of species richness and tended to be dominated by large-bodied generalists, such as *Rhinella horribilis*, *Smilisca baudinii* and *Lithobates forreri* which have wide distributions and are adapted to the seasonally xeric conditions of the dry forest. These anurans are less prone to desiccation, as their large body size means that they have proportionally lower surface area to body volume and thus lower rates of water loss than smaller bodied species (*Duellman & Trueb, 1994*) This likely explains their higher abundances and dominance in the dry forest, which is characterised by dry season high temperatures and less rain, especially during the dry season. One such adaption to the xeric conditions of the dry forest is cocoon formation, as observed in *Smilisca baudinii*, allowing them to survive long periods without rain (*McDiarmid & Foster, 1987*). The similarity between Cacao and Maritza is likely due to the proximity of these two sites (4.5 km) and that they occupy one continuous forest, albeit over an elevational gradient, rather than environmental conditions—which are grossly different between the two sites. *Duellmanohyla rufioculis* was only found at these two sites, whilst *Lithobates warszewitschii* was far more abundant in these two sites than any other.

Weather conditions at different elevations are likely to play a significant role in constraining diversity to a specific site and may explain the greater diversity found in San Gerardo compared to Cacao. For many groups of organisms, including amphibians (*Campbell, 1999*), diversity changes along an elevational gradient (*e.g.*, *McCain, 2005*; *Navas, 2003*; *Terborgh, 1971*), following a bell-shaped curve. Species richness is relatively low at lower and higher elevations, with the highest species richness recorded at mid-elevations. However, endemism in the tropics is far more ubiquitous at high elevation sites; meaning they are of great conservation priority—a consequence of these sites being far more insular (*Savage, 2002*). The results roughly follow this trend, with the average elevation of our transects in the most species rich site, San Gerardo (573.32 m), between the elevation of the less diverse higher elevation site (Cacao: 1,050.17 m) and lower elevation sites (Santa Rosa: 289.2 m, Murciélago: 80.5 m).

Despite differences in the structure of the forest habitats, two species were found to occur in all four, *Rhinella horribilis* and *Craugastor fitzingeri*. This is likely attributed to their generalist nature and ability to adapt to human altered landscapes (*Crawford, Bermingham & Carolina, 2007*). Only 11 species were found at more than one site, but some exhibited far higher abundance in only one forest type, such as *D. rufioculis* which was found at very high abundances in Cacao (131 individuals), low abundances at Maritza (three individuals) and absent from all other sites—a consequence of the elevational range constraints and climatic requirements of this species (*Savage, 2002*). Historic declines may also play a role in the presence and absence of certain species at different sites, as illustrated by *Craugastor ranoides*. This once widespread riparian species is likely highly sensitive to *B. dendrobatidis* outbreaks (known populations of this species have disappeared from most of its range in Costa Rica, and *B. dendrobatidis* was found responsible for the decline of a

highly-related species, *Craugastor punctariolus*; *Ryan, Lips & Eichholz, 2008*) and is likely only to persist in Murciélago due to the areas status as a climatic refuge, where the environmental conditions have helped prevent disease outbreaks (*Puschendorf et al., 2009*). However, this dry forest peninsula is also subject to serpentinization (*Sánchez-Murillo et al., 2014*). This produces hyperalkaline fluids, reaching a pH of >11, which drain into the local streams in which these frogs live. The potential effects of this pH change on the skin fungus and its resultant disease are yet to be explored. In Cacao forest, alongside *Craugastor ranoides, Atelopus varius, Isthmohyla tica, Craugastor andi, Duellmanohyla uranochroa* have also vanished and all salamanders are now extremely uncommon. However, more intensive sampling during different years and different seasons may reveal that these species persist, albeit in much lower numbers.

The steep decline in amphibian diversity in Cacao, over the 30-year period 1987/8–2017 is persistent and clearly recovery has been slow. A 45% reduction in species richness was observed, with only 11 species recorded in 2017 compared to 20 in the 1980's, with far greater sampling effort involved in 2017. The complete lack of salamanders on the transects was especially notable, due to their historic ubiquity in the area and this finding aligns with the declines reported by other studies on neotropical salamanders (*Acosta-Chaves et al., 2015*; *Rovito et al., 2009*). In the early 1980's and 1990's, D.H. Janzen regularly encountered salamanders under fallen, rotting tree stems (night and day) and on wet foliage at night, whilst searching for caterpillars in the vicinity of Estacioón Biológica Cacao (800–1,400 m) year-round. Since the 2000's none have been encountered by either D.H. Janzen or the parataxonomists on their daily search for caterpillars. Although we cannot say with certainty that these salamanders are locally extinct, if they are still present at Cacao it is likely at levels substantially below their pre-decline numbers and recovery to these levels appears increasingly doubtful. The historic data supports previous studies looking at herpetofauna diversity of sites at similar elevations (*Scott, 1976*: Puntarenas Province, Costa Rica). Cacao is comprised of mostly old growth forest with a few patches of forest at various stages of regeneration, which makes these declines even more alarming. But these declines match those experienced by other high elevation old growth forests in the neotropics (*Young et al., 2001*). The limited data also demonstrates that there has been little recovery of amphibian diversity following these declines. However, certain species appear to have been less affected in the long-term than others, such as *C. crassidigitus, D. rufioculis and, L. warszewitschii*, which despite experiencing similar declines, have since recovered and are now the most visible of the Cacao amphibian community. A recent study by *Acosta-Chaves et al. (2019)* found similar results with *C. crassidigitus* and *L. warszewitschii* now dominating the amphibian community of Reserva de San Ramón, despite their almost absence in the 1990s. *Voyles et al. (2018)*, examined the temporal changes in detection rates of 12 riparian species at three sites in Panama. Many of the species experienced rapid decreases during the epizootic phase of the *B. dendrobatidis* outbreak. However, following the transition to the enzootic phase, *B. dendrobatidis* prevalence decreased, concomitant to the recovery of several of the species; including *L. warszewitschii* and *C. crassidigitus*. This suggests changes in host responses to diseases.

A potential cause of these declines is the pathogenic fungus *B. dendrobatidis*, which has been reported for several frog species on Cacao (*Wynne, 2018*), although synergistic interactions among different environmental variables may conceal individual effects (*Navas & Otani, 2007*). *Scheele et al. (2019)* suggest that *B. dendrobatidis* is responsible for the decline of 501 amphibian species and the potential extinction of 90 species, making it seem to be one of the deadliest diseases for wild biodiversity. However, amphibian population collapses are not occurring in isolation—they are part of a constellation of changes taking place in tropical old growth forests (including Cacao), such as the decline of birds, lizards and insects, which are not susceptible to *B. dendrobatidis* (*Janzen & Hallwachs, 2021*; *Lister & Garcia, 2018*; *Pounds, Fogden & Campbell, 1999*; *Zipkin et al., 2020*; *Zipkin & DiRenzo, 2022*), suggesting *B. dendrobatidis* may not be the sole culprit of these observed declines. Cacao, as with many of the other regions where declines have been documented, has gone through an ecological homogenisation, with a large increase in similarity among sites following the declines (*Smith, Lips & Chase, 2009*). This is likely to be an underestimate of dissimilarity as today we know that lowland amphibian communities have also been suffering declines, just over a longer time period (*Ryan, Lips & Eichholz, 2008*; *Whitfield et al., 2007*). However, baseline data is only available for Cacao.

Documenting long-term declines is only possible through the collection of baseline data (*e.g.*, *Ryan, Lips & Eichholz, 2008*; *Whitfield et al., 2007*). The observation of a substantial decline in amphibian diversity within an old growth forest in ACG was only possible because of data collected several decades prior, by an expedition from the University of California, Berkeley. Other sites examined in this study may have experienced similar declines to that of Cacao, however we lack the data to empirically support this. ACG is in a unique position to provide a platform for understanding changes, past and present, and the resilience, or lack thereof, of tropical ecosystems and assemblages to climate change.

## ACKNOWLEDGEMENTS

A special thank you to George Coogan for helping conduct the field work. Thank you to the parataxónomos and parataxónomas and ACG staff who made us feel welcome and enabled our work throughout all the different field stations, especially Roger Blanco. Thank you to one Victor Acosta Chaves and one anonymous reviewer for feedback that improved the final version of this article.

### Funding

The authors received no funding for this work.

### Competing Interests

Xavier A. Harrison is an Academic Editor for PeerJ. Roberto Fernandez is employed by the Guanacaste Dry Forest Conservation Fund (GDFCF).

## Author Contributions

- Alex W. Edwards conceived and designed the experiments, performed the experiments, analyzed the data, prepared figures and/or tables, authored or reviewed drafts of the article, and approved the final draft.
- Xavier A. Harrison conceived and designed the experiments, analyzed the data, prepared figures and/or tables, authored or reviewed drafts of the article, and approved the final draft.
- M. Alex Smith conceived and designed the experiments, analyzed the data, prepared figures and/or tables, authored or reviewed drafts of the article, and approved the final draft.
- Maria Marta Chavarría Díaz conceived and designed the experiments, authored or reviewed drafts of the article, help coordinating permits and fieldwork, and approved the final draft.
- Mahmood Sasa conceived and designed the experiments, authored or reviewed drafts of the article, help with key background information that is not published, and approved the final draft.
- Daniel H. Janzen conceived and designed the experiments, authored or reviewed drafts of the article, and approved the final draft.
- Winnie Hallwachs conceived and designed the experiments, authored or reviewed drafts of the article, and approved the final draft.
- Gerardo Chaves conceived and designed the experiments, performed the experiments, prepared figures and/or tables, authored or reviewed drafts of the article, gave us access to historic information, and approved the final draft.
- Roberto Fernández conceived and designed the experiments, authored or reviewed drafts of the article, and approved the final draft.
- Caroline Palmer conceived and designed the experiments, authored or reviewed drafts of the article, and approved the final draft.
- Chloe Wilson conceived and designed the experiments, authored or reviewed drafts of the article, and approved the final draft.
- Alexandra North conceived and designed the experiments, authored or reviewed drafts of the article, and approved the final draft.
- Robert Puschendorf conceived and designed the experiments, performed the experiments, analyzed the data, prepared figures and/or tables, authored or reviewed drafts of the article, and approved the final draft.

## Field Study Permissions

The following information was supplied relating to field study approvals (*i.e.*, approving body and any reference numbers):

This work was carried out under CONAGEBIO permit number R-036-2013-OT-CONAGEBIO.

## Data Availability

The data is available at GitHub: https://github.com/xavharrison/CostaRica_RankAbundance_2022.

## Supplemental Information

Supplemental information for this article can be found online at http://dx.doi.org/10.7717/peerj.16185#supplemental-information.

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
