# Peer review of "Amphibian diversity across three adjacent ecosystems in Área de Conservación Guanacaste, Costa Rica"

_PeerJ, doi:10.7717/peerj.16185_

## Round 0.1 · original submission · Minor Revisions

Dear Dr. Puschendorf,

After this first review round, I believe that your manuscript may be published n PeerJ after you improve the mmanuscript according to the suggestions made by both reviewers. Please do not forget to prepare a rebuttal letter by the time you resubmit your manuscript to PeerJ.

Sincerely,
Daniel Silva, PhD

Reviewer 1 ·

Basic reporting

This manuscript is well-written and well-cited. I have included some grammatical recommendations below, but overall the language and messaging in this manuscript is on-point.
It is clear from the references cited that the authors have a deep breadth of knowledge on the topic of herpetological biodiversity, amphibian declines, and the requisite historical context. The authors adequately capture the necessary background to explain their rationale, and place the results in the relevant context.
The article is appropriately structured. The figures and tables are relevant. I have some specific recommendations on figures and tables included below. Supplementary materials include most of the data, however the raw results from surveys are not included. Many of the analyses could be replicated from the data provided.
This article is entirely self-contained and appropriately presents the information.

Experimental design

This manuscript presents original data and fits the aims and scopes of the journal.
The research question behind this study is well thought out. Gathering population and community level information on wildlife communities is crucial to our understanding of the effects of climate change, disease, and other factors. The authors identify gaps in the current knowledge and designed a study to fill those areas.
This study was rigorous and well designed given the constraints they faced. By the information presented it appears to have been done ethically and technically correct.
The methods section is sufficiently detailed to allow someone to repeat the same design for this experiment.

Validity of the findings

The authors make it clear that this study, and the data/results presented therein, is meant to be a baseline for future work.
Summary data have been provided, and all statistical tests are appropriately represented. There are no explicit controls, but they are not necessary for this observational study.
The conclusions and discussion focus on contextualizing the results and searching for potential explanations for observed diversity patterns.

Additional comments

I applaud the authors for this manuscript. It is clear that you have a strong understanding of patterns in neotropical herp diversity, the historical context of amphibian populations in Costa Rica, and the pressures facing frogs in this region. I am glad to see this research being published; baseline survey data is always valuable, particularly when considering threatened taxa. The sampling regime was well thought out, although slightly limited in scope, and the analyses are appropriate for the data and questions being asked. I have included some general thoughts here, in no particular order:
• Double check consistency of notation throughout. In some cases you use a space between a measurement and its units, and in other places you don’t. I would recommend sticking with a space (e.g., 1800 m).
• Double check grammatical styling. While I generally advocate for Oxford comma, it isn’t a necessity, but the use of commas is a bit inconsistent throughout.
• Line 412: Should be Estacion Biologica
• Line 419-422: I would recommend rewording for clarity
• Line 437: lowercase amphibian
• Generally, I would add in some more consideration of the brevity of your sampling. While you do temper your results in the discussion, it is also important to provide some context on seasonality (especially in this region). Given the short sampling season used here, it makes sense that your overall diversity estimates are much lower than what “should” be present in ACG. I don’t doubt that many of these species are still “missing” or at very low numbers, but some of the patterns you are seeing may be very attributable to seasonality and yearly fluctuations in population.
• I would also consider tempering your discussion of species recovery. While your study did not find evidence of substantial recovery, anecdotal reports do suggest that some populations have recovered (as you noted). It is certainly likely that some are lost forever, but some population recovery does seem probably – while community recovery to pre-decline levels is unlikely.
• For all figures and tables I would recommend providing more detailed captions.
• For Table 3 I recommend changing the model notation so it is easier to understand. It is not necessary to keep the notation the same as the output of the models (e.g., siteMaritza can just be Maritza)
• Table 4 may be more appropriate for the supplement. Also, it may just be formatting on the review document, but make sure your species names don’t run together.
• Figure 1 doesn’t match all the sites; San Gerardo isn’t clearly marked.
• Figure 3 should be cleaned up to match the other figures (remove gray boxes), and the labels should be Title Case
• Figure 4 – Check the spelling of your sites (Maritza)

I encourage you to continue pushing through on this manuscript. It is a worthwhile data set that is incredibly valuable as we move into a new phase of studying amphibian populations.

·

Basic reporting

This paper presents groundbreaking data from the ACG conservation area, making it an invaluable contribution to the field. The authors have skillfully crafted a well-written manuscript, showcasing their expertise in the subject matter. However, I strongly believe that the title could benefit from being more specific and informative. I have taken the liberty of suggesting some references that, in my opinion, would enhance the paper by providing additional aspects for comparison, which were initially lacking. The meticulous analysis and comprehensive data presented in this paper will serve as a pivotal reference for future research and comparative studies in the region. With some minor corrections, this paper is undoubtedly ready for publication. I urge the authors to take these suggestions into careful consideration. Heartfelt congratulations to the authors for their significant sampling effort, which undoubtedly bolsters the ongoing conservation processes in the Guanacaste region.

Experimental design

no comments

Validity of the findings

no comments

Additional comments

There is a problem with Spanish names, for example Universidad de Costa Rica, Murciélago, Río in the main text and some tables and figures, that should be corrected. Please double check Spanish words.

---

## Round 0.2 · accepted · Accept

Dear Dr. Puschendorf,

I am pleased to inform you that your manuscript has been accepted for publication in PeerJ!

I am so sorry for taking some time to review this manuscript. I was wondering whether the original reviewers could assess it, but they were unreachable. After reading your response letter and reading the whole manuscript, even though I am no expert in amphibians, I believe the text is publishable.

Best regards,
Daniel Silva, PhD